# Screen time, sleep quality and mental health among adolescents of secondary schools in Dharan

**Isha Aryal[1], Vivek Gyawali[2]\*, Nirmala Pradhan[1], Sami Lama[1], Kriti Thapa[1]**

**1** Department of Psychiatric Nursing, College of Nursing, B.P. Koirala Institute of Health Sciences, Dharan, Koshi, Nepal, **2** School of Public Health and Community Medicine, B.P. Koirala Institute of Health Sciences, Dharan, Koshi, Nepal

\* gyawalivivek19@gmail.com

## Abstract

Today's generation children and adolescents are growing up with electronic media. Although the recommended screen time is up to two hours for adolescents, the screen exceeds the recommended limit. Excessive use of screen devices has resulted in decreased sleep and had a negative impact on mental health of adolescents. The objectives of the study were to assess the screen time, sleep quality and mental health among adolescence of secondary schools in Dharan and to find out the association of screen time with socio demographic variables, sleep quality and mental health. Descriptive cross-sectional study was conducted among 259 secondary level students of Dharan. Two government and two private school were selected by simple random sampling technique. Systematic random sampling technique was used to select the respondents. Data was collected using Pittsburgh Sleep Quality Index and Patient Health Questionnaire. Data analysis was done using SPSS version 11.5. The mean screen time of respondents was 4.93±2.11 hours per day and mean sleep duration was 6.73±1.41 hours. The prevalence of poor sleep quality was 40.2% and poor mental health was 46.3%. Screen time was significantly associated with type of school (p=0.006). There was significant association of sleep quality with screen time for entertainment (p=0.002), total screen time (p=0.01) and time of maximum use (p=0.04). Mental health was significantly associated with screen time for entertainment (p=0.011) and total screen time (p=0.013). Mental health score was positively correlated with screen time. This study concludes that adolescents have higher screen time than recommended and a significant proportion of adolescents have poor sleep quality and poor mental health. Screen time is statistically significant and positively correlated with sleep quality and mental health scores.

**Data availability statement:** All data underlying the findings in our study have been fully included in the paper itself in the form of tables. The dataset used for analysis contained only the same data points presented in the paper, and no additional raw dataset exists beyond what has been reported. Since all values used for calculations (including descriptive statistics and frequencies) are fully presented in the paper, other researchers can reproduce and verify our results directly from the published data.

**Funding:** The authors received no specific funding for this work.

**Competing interests:** The authors have declared that no competing interests exist.

## Introduction

Screen time (ST) is the time spent in sedentary behaviours that involve screen-based media (SBM) such as viewing television, playing video games, and using computer and smartphones. The American Academy of Paediatrics recommended two hours or less of sedentary screen time daily for children between two to 18 years [1]. The Australian Government guidelines recommends no sedentary recreational screen time for children under two years, no more than one hour per day for two to five years and no more than two hours per day for five to eighteen years [2]. Similarly, sleep quality means getting uninterrupted and refreshing sleep which is not just about hours of sleep, but how well a person sleeps [3].

According to WHO, Mental health is a state of mental well-being that enables people to cope with the stresses of life, realize their abilities, learn well and work well, and contribute to their community [4].

In adolescents, increased exposure to a sedentary screen can affect sleep quality leading to insomnia, daytime sleepiness, impact daily activities negatively [5] and lead to depressive symptoms and decreased quality of life [6].

It is reported that 20% teenagers spend five or more hours a day on social media [7]. In adolescents, increased exposure to a sedentary screen can affect sleep quality leading to insomnia, daytime sleepiness, impact daily activities negatively [5] and lead to depressive symptoms and decreased quality of life [6].

Different studies have shown relationship between night-time mobile phone use and subsequent depressed mood, externalizing behaviour and declined self-esteem [7]. Blue spectrum light produced through various electronic devices supress production of melatonin, leading to decreased drowsiness, difficulty initiating sleep, and non-restorative sleep [8].

A study conducted in a peri-urban area of Nepal found that 31% of the adolescents who use internet excessively had poor sleep quality [9].

Globally, one in seven (14%) 10–19 years old experience mental disorder, of which depression, anxiety and behavioural disorders are most common [10]. Failure to address mental health conditions in adolescents can extend it to adulthood. A systematic review of 35 longitudinal studies showed a significant relationship between screen time and depressive symptoms [11].

Studies have shown a significant association between sleep quality and mental health related quality of life, where sleep quality accounted for approximately 17–22% of variance in mental health related quality of life [12].

A study conducted among US adolescents indicate earlier bedtimes are protective, with adolescents going to bed before 10 pm demonstrate significantly lower odds of suicidality compared to those with later bedtimes (OR = 0.59). Similarly, adolescents with sleep durations (≤7 hours) exhibited greater odds of a wide range of adverse outcomes, including mood and anxiety disorders, suicidality, substance use, behavioral disorders, tobacco smoking, and poor perceived mental health. Reported odds ratios for these associations ranged from 1.27 (95% CI, 1.06–1.54) to 2.50 (95% CI, 1.56–2.71) [13].

A critical period for achievement of social and emotional capabilities is adolescence, which becomes foundation for future life, health, and wellbeing. Sleep problems may cause psychological stress and disorders such as depression, increased suicide risk and substance abuse [14].

School going adolescents are chosen for this study as they are at a developmental stage where intersection of academic demands, peer influence, and increased exposure to digital technologies can directly affect sleep patterns and mental well-being. Focusing on in-school adolescents provides findings that are directly relevant for school-based health promotion, preventive strategies, and policy interventions, which can be implemented more feasibly within the education system through designed curriculum.

The objective of the study was to assess screen time, sleep quality and mental health among adolescents of secondary schools in Dharan. Secondary objectives were to identify the association of screen time with socio-demographic variables, sleep quality and mental health among adolescents of secondary schools in Dharan.

## Method

Ethics statement: Ethical approval was obtained from the research committee of BP Koirala Institute of Health Sciences (BPKIHS) (Reference no. 713/079/080-IRC). Then, a formal request letter for data collection was obtained from BPKIHS-CON (BP Koirala Institute of Health Sciences – College of Nursing) and written permission was obtained from the selected schools for data collection. Parental consent was obtained from parents of participants below 16 years, assents were taken from those students, and informed written consent was obtained from participants above 16 years. Privacy and confidentiality of each participant was maintained during data collection.

A descriptive cross-sectional study was used to assess the relationship of screen time with sleep quality and mental health among adolescents of secondary schools. Study was carried out in secondary schools of Dharan sub metropolitan city.

The study population comprised adolescents enrolled in secondary levels, i.e., grades 9–12. Students were eligible for inclusion if they were enrolled in grade 9–12, present on the day of data collection and willing to participate. The age range of participants was 12–19 years, corresponding to adolescents enrolled in grades 9–12. Students younger than 12 years were not encountered in the selected classes and hence were not included. Sampling was conducted in two stages. In the first stage, four schools (two government and two private) were selected using simple random sampling (lottery method) without replacement. In the second stage, students were selected proportionally from all grades (9–12) of these schools to ensure representation across levels. Class rosters served as the sampling frame, and systematic random sampling was applied within each class. The sampling interval (k) was calculated by dividing the total number of students in a class by the number of participants required, and a random starting point was chosen by lottery. Thereafter, every kth student was selected until the required number of participants was reached.

The required sample size was determined using the formula $(n) = \frac{Z^2PQ}{d^2}$, where Z = 1.96 at 95% confidence level, P = 29%, Q = (100% - P) =71% based on a previous study from Vastmanland, Sweden, which reported a 29% prevalence of mental health problems associated with high screen time [15]. The allowable error (d) was set at 20% of P (=0.058). This yielded a minimum sample size of 235. To account for possible non-response, the 10% of sample size was added resulting in a final required sample of 259 participants.

Outcome variables of the study were mental health, sleep quality whereas explanatory variable were screen time, socio-demographic factors (age, gender, grade, type of family, type of school, occupation of parents, history of medical illness in family, history of mental illness in family) and factors related to use (access to devices, availability of internet at home, purpose of use).

A set of self-administered questionnaires was used which includes Socio-demographic information, Screen Time Questionnaire (STQ), Pittsburgh Sleep Quality Index (PSQI) and Patient Health Questionnaire (PHQ-9). Pittsburgh Sleep Quality Index (PSQI) is highly reliable and valid instrument to measure the quality and pattern of sleep. It has 7 internal

components, and each component is scored as 0–3. The component scores are summed to produce a global score range of 0–21 where high scores indicate worse quality sleep. Scores between 0–5 were considered as 'good sleep quality' and ≥6 as 'poor sleep quality' [9]. Patient Health Questionnaire (PHQ-9) is a 9-item self-administered tool, that is used to assess depression in adolescents. The score of each item ranges from 0 (not at all) to 3 (almost every day) based on frequency of symptoms experienced. The score ranges from 0 to 27 where a greater score (≥10) reflects depression [16].

Pretesting was done among 10% of the sample size in similar setting to identify feasibility, completeness, comprehensiveness, and appropriateness before actual data collection. The population of pretest were excluded in the actual study. Necessary modification such as specifying type of mental illness in family was done after the pretesting.

After data collection, collected data were checked for completeness, organized, coded and entered in Microsoft Excel and converted into SPSS version 11.5. Older version of SPSS was used as it is the only licensed version of the product that was available to the researchers. Descriptive statistics (frequency, percentage, mean and standard deviation) were calculated for socio demographic variables. Then some characteristics (ethnicity, religion, grade, parent's occupation, and screen time) were dichotomized for calculating inferential statistics. Inferential statistics Pearson Chi Square test and Fisher's exact test was applied to find out the significant association of sociodemographic variables with screen time and mental health as well as screen time variables with sleep quality and mental health. Spearman's correlation coefficient was used to correlate between screen time and sleep quality score, and screen time and mental health score. For entire test, confidence interval was considered 95% (p = 0.05).

## Results

Table 1 shows that mean age of the respondents was 16.01 ± 1.21 years. Majority (67.6%) of the respondents were from age group 16–19 years. More than half (57.5%) of the respondents were female. Majority (74.5%) of them belonged to Janajati by ethnicity. Nearly half (42.5%) of them were Hindu by religion, followed by 32.8% Kirat religion. Most of the respondents (27.4%) were from class 11. Half of the respondents (50.2%) were from government schools. Almost half (50.6%) of them were from nuclear family. Majority of the respondents (93%) had both parents. Most of the respondents' (33.3%) father were foreign employed. More than half (55.0%) of the respondents' mother were homemaker Almost one third (32%) of respondents had medical illness in their family and 7.3% of respondents had mental illness in their family.

Similarly, Table 2 depicts that all the respondents (100%) had access to screen-based devices. More than half (51.7%) of the respondents used the devices at night. Also more than one quarter (28.6%) of the respondents were using screen in the afternoon as they were morning students. Majority (95%) of them had access to internet. Majority (63.7%) of the respondents used the devices less than two hours for education and 36.3% used the devices for entertainment for two to four hours per day. Majority (35.1%) of the respondents' total screen time was four to six hours. Only 5.4% use screen less than two hours per day. The mean screen time of the respondents was 4.93 ± 2.11 hours per day.

Likewise, Table 3 reflects that less than half (40.2%) of the respondents had poor sleep quality. The mean sleep duration of respondents was 6.73 ± 1.41 hours per day.

Now, Table 4 shows mental health status of the respondents which depicts that almost half (46.3%) of the respondents had depression.

Here, Table 5 represents about the association between screen time and demographic variable where, screen time was found to be significantly associated with type of school.
Similarly, Table 6 presents the association between mental health status and selected socio-demographic variables of the respondents. The analysis revealed a statistically significant association between depression and several factors including gender, type of school attended, history of medical illness in the family, and history of mental illness in the family.
Likewise, Table 7 illustrates the association between screen time variables and the sleep quality of respondents. The cut off points for screen time was taken as per the recommendation from The American Academy of Paediatrics which recommends two hours or less of sedentary screen time daily for children between two to 18 years [1]. The analysis indicates a

**Table 1. Socio demographic characteristics of respondents (n = 259).**

| Characteristics | Categories | Frequency | Percentage |
|---|---|---|---|
| Age in years | 12-15 | 84 | 32.4 |
| | 16-19 | 175 | 67.6 |
| Mean age in years ± SD = 16.01 ± 1.21 | | | |
| Gender | Female | 149 | 57.5 |
| | Male | 110 | 42.5 |
| Ethnicity | Janajati | 193 | 74.5 |
| | Dalit | 30 | 11.6 |
| | Brahmin/Chhetri | 25 | 9.6 |
| | Madhesi | 9 | 3.5 |
| | Muslim | 2 | 0.8 |
| Religion | Hindu | 110 | 42.5 |
| | Kirat | 85 | 32.8 |
| | Buddhist | 39 | 15.0 |
| | Christian | 23 | 8.9 |
| | Islam | 2 | 0.8 |
| Grade | 9 | 61 | 23.6 |
| | 10 | 58 | 22.4 |
| | 11 | 71 | 27.4 |
| | 12 | 69 | 26.6 |
| Type of school | Government | 130 | 50.2 |
| | Private | 129 | 49.8 |
| Type of family | Nuclear | 131 | 50.6 |
| | Joint | 128 | 49.4 |
| Parental Status | Presence of both parents | 242 | 93.4 |
| | Single parent | 16 | 6.2 |
| | Absence of both parents | 1 | 0.4 |
| Father's occupation | Foreign Employment | 83 | 32.0 |
| | Agriculture | 65 | 25.1 |
| | Business | 44 | 17.0 |
| | Service | 41 | 15.8 |
| | Painter and Labor | 11 | 4.2 |
| | Unemployed | 5 | 1.9 |
| | No father | 10 | 3.9 |
| Mother's occupation | Homemaker | 138 | 53.3 |
| | Agriculture | 57 | 22.0 |
| | Business | 21 | 8.1 |
| | Foreign employment | 20 | 7.7 |
| | Service | 13 | 5.0 |
| | Labor | 2 | 0.8 |
| | No mother | 8 | 3.1 |
| History of medical illness in family | Yes | 83 | 32.0 |
| | No | 176 | 68.0 |
| History of mental illness in family | Yes | 19 | 7.3 |
| | No | 240 | 92.7 |

**Table 2. Frequency and percentage on screen time questionnaire of respondents (n = 259).**

| Characteristics | Categories | Frequency | Percentage |
|---|---|---|---|
| Access to Screen Based Devices | Yes | 259 | 100 |
| | No | 0 | 0 |
| Time of maximum use | Night | 134 | 51.7 |
| | Afternoon | 74 | 28.6 |
| | Evening | 47 | 18.2 |
| | Morning | 4 | 1.5 |
| Access to internet | Yes | 246 | 95.0 |
| | No | 13 | 5.0 |
| Screen time for education (hours per day) | <2 hours | 165 | 63.7 |
| | 2-4 hours | 90 | 34.7 |
| | 4-6 hours | 4 | 1.6 |
| Mean ST for Education in hours ± SD = 1.44 ± 0.96 | | | |
| Screen time for entertainment (hours per day) | <2 hours | 57 | 22.0 |
| | 2-4 hours | 94 | 36.3 |
| | 4-6 hours | 74 | 28.6 |
| | 6-8 hours | 24 | 9.2 |
| | 8-10 hours | 9 | 3.5 |
| | >10 hours | 1 | 0.4 |
| Mean ST for Entertainment in hours ± SD = 3.49 ± 1.98 | | | |
| Total Screen Time (hours per day) | <2 hours | 14 | 5.4 |
| | 2-4 hours | 71 | 27.4 |
| | 4-6 hours | 91 | 35.1 |
| | 6-8 hours | 54 | 20.9 |
| | 8-10 hours | 24 | 9.3 |
| | >10 hours | 5 | 1.9 |
| Mean Total Screen Time in hours ± SD = 4.93 ± 2.11 | | | |

**Table 3. Sleep quality of respondents (n = 259).**

| Characteristics | Frequency | Percentage |
|---|---|---|
| Good Sleep Quality (PSQI <6) | 155 | 59.8 |
| Poor Sleep Quality (PSQI ≥6) | 104 | 40.2 |
| Mean sleep duration in hours: 6.73 ± 1.41 | | |

**Table 4. Mental health status of respondents (n = 259).**

| Characteristics | Frequency | Percentage |
|---|---|---|
| No depression (PHQ < 10) | 139 | 53.7 |
| Depression (PHQ ≥ 10) | 120 | 46.3 |

statistically significant association between sleep quality and screen time for entertainment, total screen time, and the time of maximum screen use. These results highlight the potential impact of digital media habits on sleep health among the study population.

**Table 5. Association between screen time and demographic variables of respondents (n = 259).**

| Characteristics | Categories | Screen time | | P value | Odds Ratio# (95% CI) |
|---|---|---|---|---|---|
| | | >2 hours | <2 hours | | |
| Age in years | 12-15 | 81 (95.3%) | 3 (4.7%) | 0.558** | 1.811 (0.492-6.672) |
| | 16-19 | 164 (93.7%) | 11 (6.3%) | | |
| Gender | Female | 144 (96.6%) | 5 (3.4%) | 0.090* | 2.566 (0.835-7.885) |
| | Male | 101 (91.8%) | 9 (8.2%) | | |
| Ethnicity | Janajati | 182 (94.3%) | 11 (5.7%) | >0.999** | 0.788 (0.213-2.915) |
| | Others | 63 (95.5%) | 3 (4.5%) | | |
| Religion | Hindu | 103 (93.6%) | 7 (6.4%) | 0.558* | 0.725 (0.247-2.131) |
| | Others | 142 (95.3%) | 7 (4.7%) | | |
| Grade | 9 and 10 | 112 (94.1%) | 7 (6.9%) | 0.754* | 0.842 (0.287-2.473) |
| | 11 and 12 | 133 (95.0%) | 7 (5.0%) | | |
| Type of school | Government | 118 (90.8%) | 12 (9.2%) | **0.006*** | 0.155 (0.034-0.706) |
| | Private | 127 (98.4%) | 2 (1.6%) | | |
| Type of family | Nuclear | 125 (95.4%) | 6 (4.6%) | 0.552* | 1.389 (0.468-4.122) |
| | Joint | 120 (93.8%) | 8 (6.2%) | | |
| Father's occupation | Service | 40 (97.6%) | 1 (2.4%) | 0.477** | 2.887 (0.369-22.583) |
| | Others | 194 (93.3%) | 14 (6.7%) | | |
| Mother's occupation | Homemaker | 134 (97.1%) | 4 (2.9%) | 0.053** | 3.252 (0.992-10.666) |
| | Working Mother | 103 (91.1%) | 10 (8.9%) | | |
| Medical illness in family | No | 166 (94.3%) | 10 (5.7%) | >0.999** | 0.841 (0.256-2.763) |
| | Yes | 79 (95.2%) | 4 (4.8%) | | |
| Mental illness in family | No | 227 (94.6%) | 13 (5.4%) | >0.999** | 0.979 (0.120-7.842) |
| | Yes | 18 (94.7%) | 1 (5.3%) | | |

\* Pearson chi square test ** Fisher Exact Test # Unadjusted odds ratio.

Also, Table 8 presents the association between screen time variables and the mental health status of respondents. The findings show a statistically significant association between mental health and both screen time for entertainment and total screen time.

Lastly, Table 9 depicts the correlation between screen time and both sleep quality and mental health scores. The analysis reveals a positive correlation, with screen time significantly associated with higher sleep quality and mental health scores (p<0.001).

## Discussion

Present study showed few (5.4%) respondents used screen devices as recommended, that is less than two hours per day. This finding is comparable with a study from India, where 17% respondents had screen time as recommended [17]. The finding is in contrast with the study conducted in China, which showed 27.3% respondents used screen time as recommended. It may be contradicting due to the large sample size of Chinese study [18].

In the present study, mean screen time of adolescents was 4.93±2.11 hours, which is consistent with the findings from a study conducted in Brazil which also showed the median screen time to be 4.5 hours [19]. This may be due to wide technological advancement and dependence on screen devices for academic and recreational purpose by adolescence of today's generation.

**Table 6. Association between mental health status and socio-demographic variables of respondents (n = 259).**

| Characteristics | Categories | Mental Health Status | | P value | Odd's Ratio# (95% CI) |
|---|---|---|---|---|---|
| | | Depression | No depression | | |
| Age in years | 12-15 | 36 (42.9%) | 48 (57.1%) | 0.437* | 0.813 (0.481-1.372) |
| | 16-19 | 84 (48.0%) | 91 (52.0%) | | |
| Gender | Female | 82 (55.0%) | 67 (45.0%) | **0.001*** | 2.319 (1.395-3.856) |
| | Male | 38 (34.5%) | 72 (65.5%) | | |
| Ethnicity | Janajati | 87 (45.1%) | 106 (54.9%) | 0.489* | 0.821 (0.469-1.436) |
| | Others | 33 (50.0%) | 33 (50.0%) | | |
| Religion | Hindu | 47 (42.7%) | 63 (57.3%) | 0.317* | 0.777 (0.473-1.275) |
| | Others | 73 (49.0%) | 76 (51.0%) | | |
| Grade | 9 and 10 | 59 (49.6%) | 60 (50.4%) | 0.334* | 1.273 (0.780-2.080) |
| | 11 and 12 | 61 (43,6%) | 79 (56.4%) | | |
| Type of school | Government | 33 (25.4%) | 97 (74.6%) | **<0.001*** | 0.164 (0.96-0.282) |
| | Private | 87 (67.4%) | 42 (32.6%) | | |
| Type of family | Nuclear | 68 (51.9%) | 63 (48.1%) | 0.069* | 1.578 (0.965-2.580) |
| | Joint | 52 (40.6%) | 76 (59.4%) | | |
| Father's occupation | Service | 24 (58.5%) | 17 (41.5%) | 0.083* | 1.815 (0.920-3.579) |
| | Others | 91 (43.8%) | 117 (56.2%) | | |
| Mother's occupation | Homemaker | 66 (47.8%) | 72 (52.2%) | 0.775* | 1.075 (0.653-1.770) |
| | Working | 52 (46.0%) | 61 (54.0%) | | |
| Medical illness in family | No | 73 (41.4%) | 103 (58.6%) | **0.023*** | 0.543 (0.320-0.920) |
| | Yes | 47 (56.6%) | 36 (43.4%) | | |
| Mental illness in family | No | 106 (44.2%) | 134 (55.8%) | **0.013*** | 0.283 (0.099-0.809) |
| | Yes | 14 (73.7%) | 5 (26.3%) | | |

* Pearson chi square test ** Fisher Exact Test # Unadjusted odds ratio.

**Table 7. Association of screen time variables with sleep quality of respondents (n = 259).**

| Characteristics | Categories | Sleep Quality | | P value | Odd's Ratio # (95% CI) |
|---|---|---|---|---|---|
| | | Poor | Good | | |
| Screen time for education | >2 hours | 39 (41.5%) | 55 (59.6%) | 0.741* | 1.091 (0.651-1.827) |
| | <2 hours | 65 (39.4%) | 100 (60.6%) | | |
| Screen time for entertainment | >2 hours | 91 (45.0%) | 111 (55.0%) | **0.002*** | 2.775 (1.409-5.466) |
| | <2 hours | 13 (22.8%) | 44 (77.2%) | | |
| Total Screen Time | >2 hours | 103 (42.0) | 142 (58.0%) | **0.010*** | 9.430 (1.214-73.227) |
| | <2 hours | 1 (0.7%) | 13 (99.3%) | | |
| Access to internet | Yes | 2 (15.4%) | 11 (84.6%) | 0.062* | 0.257 (0.056-1.1183) |
| | No | 102 (41.5%) | 144 (58.5%) | | |
| Time of maximum use | Morning to Afternoon | 21 (26.9%) | 57 (73.1%) | **0.040*** | 0.435 (0.225-0.777) |
| | Evening to Night | 83 (45.9%) | 98 (54.1%) | | |

*Pearson chi square test; # Unadjusted odds ratio.

A cross-sectional study conducted by Shrestha R, et.al., showed that participants from private schools were two times more likely to have high screen time than those from public schools [20]. This finding is consistent with present study which showed students from private schools are more likely to have high screen time.

**Table 8. Association of screen time variables with mental health of respondents (n = 259).**

| Characteristics | Categories | Mental Health Status | | P value | Odd's Ratio# (95% CI) |
|---|---|---|---|---|---|
| | | Depression | No depression | | |
| Screen time for education | >2 hours | 37 (44.0%) | 57 (56.0%) | 0.090* | 0.641 (00.384-1.072) |
| | <2 hours | 83 (50.3%) | 82 (49.7%) | | |
| Screen time for entertainment | >2 hours | 102 (49.5%) | 100 (50.5%) | **0.011*** | 2.210 (1.185-4.120) |
| | <2 hours | 18 (31.6%) | 39 (68.4%) | | |
| Total Screen Time | >2 hours | 118 (48.2%) | 127 (51.8%) | **0.013*** | 5.575 (1.222-25.433) |
| | <2 hours | 2 (14.3%) | 12 (85.7%) | | |
| Access to internet | No | 5 (38.5%) | 8 (61.5%) | 0.559* | 0.712 (0.027-2.237) |
| | Yes | 115 (46.7%) | 131 (53.3%) | | |
| Time of maximum use | Morning to Afternoon | 30 (38.5%) | 48 (61.5%) | 0.095* | 0.632 (0.368-1.086) |
| | Evening to Night | 90 (49.7%) | 91 (50.3%) | | |

* Pearson chi square test # Unadjusted odds ratio.

**Table 9. Correlation of screen time with sleep quality and mental health score.**

| Characteristics | Sleep Quality Score | | Mental Health Score | |
|---|---|---|---|---|
| | r | P value | r | P value |
| **Screen Time** | 0.320* | <0.001 | 0.241 | <0.001* |

* Spearman rho correlation.

Present study revealed majority (59.8%) respondents had good sleep quality, which is consistent with a study conducted among adolescents of India, where 52.5% adolescents had good sleep quality [17]. This result is comparable with study in Nepal, where 69% of the respondents had good sleep quality [9].

In this study, mean sleep duration was 6.73 ± 1.41 hours, which is consistent with a study from Brazil, where the mean sleep duration among adolescents was 7.19 hours [19]. It is also consistent with another cross-sectional study conducted in Karachi, according to which, mean total sleep duration was 6.7 ± 1.5 hours [21].

A study done in Sarlahi showed that 57.7% secondary level students were depressed, which is somehow consistent with this study where, almost half 46.3% of respondents had poor mental health. This may be because of presence of high psychological distress among adolescents of that age group [22].

Present study showed, females (55.0%) were 2.319 times more likely to have poor mental health as compared to males (34.5%), which is consistent with the study from Sarlahi and a study conducted in Birtamod [23]. This might be because females face hormonal changes and are susceptible to face psychosocial stressors. Studies have also shown that females tend to ruminate more and are also likely to seek help than the male counterparts [22].

A conducted by Giri R, et.al., also showed that students in private schools are at higher odds of experiencing depressive symptoms [23]. This finding is consistent with the current study where more students from private schools have depression. Adolescents in private schools may be more prone to experiencing depressive symptoms as they often encounter higher academic demands, limited leisure time, social comparison and intensified peer competition, which together elevate stress compared to those in public schools.

In the present study, duration of screen time was significantly associated ($p = 0.013$) with mental health, which is consistent with the findings from the study in Canada ($p = 0.001$). This may be because spending more time on screens can lead to comparison with others, social isolation and decreased time spent outdoors, which are some of the factors leading to poor mental health [24].

A study among Chinese students showed that high screen time was significantly positively associated with depression, which is consistent with the present findings, where the respondents having high screen time of more than two hours per day are 5.57 times more likely to have depression [18]. This finding is also consistent with a longitudinal study from Canada which showed that every one hour spent in screen devices cause 0.64 unit increase in depressive symptoms [25] and a study conducted by Twenge JM, et.al., in US showed that users having high screen time were more likely to have anxiety or depression [26].

The present study reveals that respondents having high screen time were 9.43 times more likely to have poor sleep quality, which is in line with the cohort study from Brazil which has revealed that increase in screen time is associated with reduction of sleep. The relation between screen time with sleep duration and sleep quality might be because as adolescents spend more time on screen devices, it substitute the time spent on other activities including sleeping [19].

In the present study, screen time was positively correlated with mental health score, which is consistent with the result of study from Brazil ($r = 0.393$) [27]. This result is also consistent with the findings from Chinese study, where screen time of more than two hours per day were positively correlated with psychological symptoms [28]. This might be because spending more time on screen based devices may expose individual to content that induce stress, disrupt sleep pattern and lead to fatigue, which contribute to mental health challenges.

The study findings align with recent evidence which highlights the importance of integrating digital and interactive approaches to promote adolescent health literacy and mental well-being [29].

In Nepal, a setting with limited prior research, this study adds to existing knowledge by evidencing the link between screen time, sleep quality, and adolescent mental health. The findings highlight the need for integrating adolescent mental health into national policies, promoting healthy sleep and digital habits through school-based interventions, and guiding parents, teachers and health professionals in early prevention and management strategies.

## Conclusion

This study concludes that most of the adolescents studying at secondary level from selected schools in Dharan have screen time higher than recommended for their age. The study also revealed that many students have poor sleep quality and poor mental health. Screen time is higher in students at private schools. Likewise, poor mental health is associated with gender and type of school. Similarly, students having high screen time have poor sleep quality and poor mental health. The results indicate that screen time have significant positive correlation between sleep quality score and mental health score. These findings highlight the need for preventive measures, awareness programs, and focused interventions by parents, schools, and policymakers to foster healthy sleep, balanced screen use, and improved mental health among adolescents.

## Limitations

The present study has several limitations. First, its cross-sectional design does not allow establishing causal relationships between screen time, sleep quality, and mental health. Second, the study was conducted in a single city, which may limit

the generalizability of the findings. Third, the use of self-reported questionnaires may introduce recall bias and social desirability bias.

## Acknowledgments

The author sincerely acknowledges the academic institution, research supervisors, and faculty members for their insightful guidance and unwavering support throughout the course of the study. The contribution of the Institutional Review Committee in granting ethical approval is also gratefully recognized. Appreciation is extended to the participating schools and study respondents for their cooperation, as well as to those who assisted in data collection, statistical analysis, and literature review. The author also extends heartfelt thanks to family and peers for their continued encouragement and support.

## Author contributions

**Conceptualization:** Isha Aryal.

**Data curation:** Isha Aryal, Vivek Gyawali.

**Formal analysis:** Isha Aryal, Vivek Gyawali.

**Investigation:** Isha Aryal.

**Methodology:** Isha Aryal, Kriti Thapa.

**Resources:** Isha Aryal.

**Supervision:** Vivek Gyawali, Nirmala Pradhan, Sami Lama, Kriti Thapa.

**Writing – original draft:** Isha Aryal.

**Writing – review & editing:** Vivek Gyawali, Nirmala Pradhan, Sami Lama, Kriti Thapa.

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
