## [Decision Letter · Decision Letter 0]

9 Sep 2025

PMEN-D-25-00306

Screen time, sleep quality and mental health among adolescents of secondary schools in Dharan

PLOS Mental Health

Dear Dr. Vivek Gyawali,

Thank you for submitting your manuscript to PLOS Mental Health. After careful consideration, we feel that it has merit but does not fully meet PLOS Mental Health’s publication criteria as it currently stands. Therefore, we invite you to submit a revised version of the manuscript that addresses the points raised during the review process. 

We look forward to receiving your revised manuscript.

Kind regards,

Olumide Thomas Adeleke, MBBS, FWACP

Academic Editor

PLOS Mental Health

Journal Requirements:

1. In the online submission form, you indicated that The datasets generated and analyzed during the current study are available from the corresponding author upon reasonable request. Due to ethical restrictions and the presence of sensitive personal information related to adolescent mental health, data access is limited to protect participant confidentiality. 

3. Uploaded as supplementary information.

Reviewers' comments:

Reviewer's Responses to Questions

**Comments to the Author**

1. Does this manuscript meet PLOS Mental Health’s publication criteria? Is the manuscript technically sound, and do the data support the conclusions? The manuscript must describe methodologically and ethically rigorous research with conclusions that are appropriately drawn based on the data presented.

Reviewer #1: Yes

Reviewer #2: Yes

Reviewer #3: Yes

2. Has the statistical analysis been performed appropriately and rigorously?

Reviewer #1: Yes

Reviewer #2: Yes

Reviewer #3: Yes

3. Have the authors made all data underlying the findings in their manuscript fully available (please refer to the Data Availability Statement at the start of the manuscript PDF file)?

Reviewer #1: No

Reviewer #2: Yes

Reviewer #3: Yes

4. Is the manuscript presented in an intelligible fashion and written in standard English?

Reviewer #1: Yes

Reviewer #2: Yes

Reviewer #3: Yes

5. Review Comments to the Author

Reviewer #1: Introduction

The introduction is too small

The author should define mental health

The author should define sleep

Author should discuss how sleep could relate to mental health

The author should discuss mental health and how it relates to sleep

The authors should discuss previous studies finding on screen time and

The gap the study is filling should be started clearly

The burden of the problem is not clearly written in the introduction

Method

Eligibility criteria should be stated

The sampling technique should state clearly. How were the students selected from the identified secondary schools?

The author should specify the age range. Is it all adolescents age 10-19years that were included in the study?

Did the author select students from all the classes in secondary school?

What is the main outcome variable and what is/are the explanatory variables in the manuscript? The author should state this under measurement of variables in the methodology.

The study instruments should be described and referenced

Results

Why is the age range starting from 12years? Adolescents start from 10-19 years according to World Health Organisation(WHO). Were some adolescents excluded? if yes the author should state this clearly

Kindly explain “as they were morning students” were some students attending afternoon school

Discussion

The author should explain why the adolescents in private schools’ experience depressive symptoms compared to their counterpart in Public schools

Reviewer #2: I sincerely thank the editor and the authors for giving me the opportunity to review the manuscript entitled “Screen time, sleep quality and mental health among adolescents of secondary schools in Dharan” (PMEN-D-25-00306). The topic addressed is highly relevant and timely, considering the increasing impact of prolonged digital device use on adolescents’ mental health and sleep quality. The study has several strengths: the sample size of 259 students is adequate, the use of validated instruments such as the Pittsburgh Sleep Quality Index (PSQI) and the Patient Health Questionnaire (PHQ-9) enhances methodological robustness, and the statistical analysis is comprehensive and well-structured, including appropriate tests and detailed correlations. However, there are several areas where clarifications, improvements, and integrations are necessary to strengthen the manuscript and increase its scientific impact.

First, the abstract should be made clearer and more informative: at line 14, the sentence “has resulted decreased sleep” should be revised to “has resulted in decreased sleep and had a negative impact”; furthermore, it would be beneficial to include the main p-values supporting the key findings to strengthen the conclusions. At line 50, there is a typo in the “Key words” section: “screen quality” should be corrected to “sleep quality”.

The introduction is well-structured but should be expanded to provide a more comprehensive background. At line 75, where it states “20% teenagers spend five or more hours a day on social media,” an appropriate reference should be added. Moreover, I recommend adding a short paragraph explaining the physiological mechanisms through which prolonged screen exposure affects sleep quality and mental health, citing recent literature on blue light effects, melatonin suppression, and circadian rhythm disruption.

In the methodology section (lines 125-185), the study is generally well-organized, but certain elements require more transparency. At line 150, where “systematic random sampling” is mentioned, please specify the k-value and starting point used to ensure replicability. At line 165, clarify the cut-off thresholds adopted to define “poor sleep quality” and “poor mental health” rather than requiring readers to infer them from the tables. Additionally, please justify the use of SPSS v11.5, which is an outdated version, or specify why it was selected for this analysis.

The results section is presented clearly, but a few improvements would enhance readability. In Table 1 (line 230), I suggest standardizing the structure of the ethnic categories for clarity. In Table 5 (line 350), please clarify whether the reported odds ratios are adjusted or unadjusted. In Tables 7 and 8, explicitly state the cut-off points defining “high” and “low” screen exposure. To improve conciseness, I also recommend removing some repetitive sentences between the results descriptions and the data already presented in the tables (lines 380-400).

The discussion is one of the strengths of this manuscript, as it effectively compares the findings with the international literature. However, several enhancements could improve its coherence and depth. At line 640, I recommend adding a brief reflection on potential sampling biases, such as the exclusion of absent students or those with undiagnosed disorders, which may have influenced the results. At line 700, please expand the comparison with international studies, clearly highlighting methodological and contextual differences that might explain variations in findings. Moreover, I suggest devoting more attention to discussing the long-term consequences of excessive screen time on adolescent mental health, supported by recent studies.

A critical omission in the manuscript is the lack of a dedicated limitations section. I strongly recommend adding a paragraph at the end of the discussion, for example:

“The present study has several limitations. First, its cross-sectional design does not allow establishing causal relationships between screen time, sleep quality, and mental health. Second, the study was conducted in a single city, which may limit the generalizability of the findings. Third, the use of self-reported questionnaires may introduce recall bias and social desirability bias.”

In the conclusion (line 810), correct the typo “scree time” to “screen time” and add a short statement on the practical implications of the findings, suggesting prevention strategies, awareness campaigns, and targeted interventions for parents, schools, and policymakers.

Finally, I strongly recommend integrating the following article into the discussion, as it provides up-to-date and highly relevant insights on digital and interactive approaches to improving adolescent mental health and health literacy:

Mancone, S., Corrado, S., Tosti, B., Spica, G., & Diotaiuti, P. (2024). Integrating digital and interactive approaches in adolescent health literacy: a comprehensive review. Frontiers in Public Health, 12, 1387874. https://doi.org/10.3389/fpubh.2024.1387874

I suggest citing it around line 730, immediately after the comparison with international literature, with a sentence such as:

“These findings align with recent evidence highlighting the importance of integrating digital and interactive approaches to promote adolescent health literacy and mental well-being (Mancone et al., 2024).”

In summary, this manuscript addresses a highly relevant topic and presents important findings, but it requires significant revision to improve its clarity, completeness, and scientific rigor. With the recommended changes, the study could make a valuable contribution to the literature on the relationship between screen time, sleep quality, and mental health among adolescents.

Reviewer #3: There is a need to revise the abstract for grammatical syntax and concord. There is a need to insert prepositions in the second and third lines of the background sub-section. In the material and methods subsection kindly insert 'a' before the descriptive cross-sectional study and delete the word 'design'.

Introduction: The concept of screen time was introduced but not adequately cited. It is important to provide at least an additional definition for excessive and normal/healthy screen time with appropriate references. The burden of prolonged or unhealthy screen time globally,in Asia and countries in sub-Saharan Africa may be mentioned to reflect the problem statement in addition to the study cited from Nepal. It is important to define what the authors mean (operational definition) by sleep quality and mental health. How do these three concepts concern adolescents? What is the justification for carrying out this study among in-school adolescents (as opposed to other youth) in Nepal?

Method

The word 'design' should be deleted since it's a descriptive study that was carried out and not a design. A brief description of the study area may be added to give more insight to the social and geographic features of the students' environment. The eligibility criteria were not clearly stated. I wish to recommend a revision of the sample size estimation because the prevalence quoted in the equation wasn't clear. Is it the prevalence of prolonged screen time or poor sleep quality from a previous study? Calculation of 10% non-response simply involves dividing the estimated sample size by 0.9.

The sampling technique will benefit from some revision. it seems multi-stage sampling was carried out. Was any form of stratification done? How were various levels (years) of study represented? What is the meaning of BPKIHS-CON? IT is of great importance to describe the study instrument to an extent where there are no ambiguities as to its content. The instrument contains three standardized (validated) tools which were not described nor referenced.Regarding data analysis, how was each of the variables of interest measured? How did each category arise? Kindly describe the scoring system.briefly. I believe there are available literature outlining the scoring guidelines.

Results

The criteria for good sleep quality and depression should have been clearly stated in the methods section. On the table, screen time may as well be represented as healthy/normal and excessive.

Discussion: kindly highlight the contribution of this paper to knowledge and its implication for policy analysis and formulation, management and practice.

6. PLOS authors have the option to publish the peer review history of their article (what does this mean?). If published, this will include your full peer review and any attached files.

**Do you want your identity to be public for this peer review?** For information about this choice, including consent withdrawal, please see our Privacy Policy.

Reviewer #1: **Yes: **Funmito Omolola Fehintola

Reviewer #2: No

Reviewer #3: No

---

## [Editor Report · Decision Letter 1]

23 Oct 2025

Screen time, sleep quality and mental health among adolescents of secondary schools in Dharan

PMEN-D-25-00306R1

Dear Vivek Gyawali,

We are pleased to inform you that your manuscript 'Screen time, sleep quality and mental health among adolescents of secondary schools in Dharan' has been provisionally accepted for publication in PLOS Mental Health.

Best regards,

Olumide Thomas Adeleke, MBBS, FWACP

Academic Editor

PLOS Mental Health